# Analyzing Fairness of Neural Network Prediction via Counterfactual Dataset Generation

Brian Hyeongseok Kim[*1], Jacqueline L. Mitchell[1], and Chao Wang[1]

[1]University of Southern California
{brian.hs.kim, jlm41510, wang626}@usc.edu

## Abstract

Interpreting the inference-time behavior of deep neural networks remains a challenging problem. Existing approaches to counterfactual explanation typically ask: What is the closest alternative *input* that would alter the model's prediction in a desired way? In contrast, we explore **counterfactual datasets**. Rather than perturbing the input, our method efficiently finds the closest alternative *training dataset*, one that differs from the original dataset by changing a few labels. Training a new model on this altered dataset can then lead to a different prediction of a given test instance. This perspective provides a new way to assess fairness by directly analyzing the influence of label bias on training and inference. Our approach can be characterized as probing whether a given prediction depends on biased labels. Since exhaustively enumerating all possible alternate datasets is infeasible, we develop analysis techniques that trace how bias in the training data may propagate through the learning algorithm to the trained network. Our method heuristically ranks and modifies the labels of a bounded number of training examples to construct a counterfactual dataset, retrains the model, and checks whether its prediction on a chosen test case changes. We evaluate our approach on feedforward neural networks across over 1100 test cases from 7 widely-used fairness datasets. Results show that it modifies only a small subset of training labels, highlighting its ability to pinpoint the critical training examples that drive prediction changes. Finally, we demonstrate how our counterfactual datasets reveal connections between training examples and test cases, offering an interpretable way to probe dataset bias.

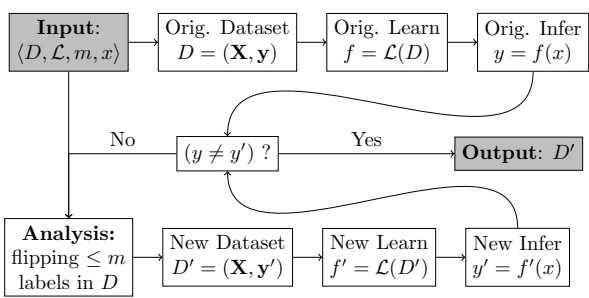

**Figure 1.** Our method to efficiently generate counterfactual dataset (CFD) $D'$ for an input $x$, based on the original dataset $D$, learning algorithm $\mathcal{L}$, and a bound $m$ on the number of training label flips.

## 1 Introduction

As machine learning models are widely deployed in socially sensitive domains such as healthcare, finance, and public policy [1–5], reasoning about their fairness has become critical. For example, an individual predicted as low-income may suspect that the model outcome reflects bias against their protected attribute (e.g., sex or race). A common way to address this problem is through inference-only methods. For example, one can perform fairness testing [6–11] and verification [12–16]. These approaches evaluate whether a trained model treats similar individuals similarly according to various fairness definitions [17–19]. Another way is through counterfactual explanations [20–24], which examine whether the trained model's prediction would change if some of the individual's attributes were altered.

Our work takes a step further by examining the training dataset itself. It is well-known that bias in training data, arising from human subjectivity or historical prejudices, can lead to unfair predictions [25]. While prior work has examined dataset bias by flipping sensitive attributes across the entire dataset to generate an alternative model and then perform fairness adjustments [26], our focus is on the *labels* instead. By selectively modifying labels to construct counterfactual datasets, we retrain a slightly altered model and check whether its prediction for the same individual changes, thereby auditing how the original decision may have been influenced by label bias [27, 28]. While this may resemble noisy label correction, our goal is not to improve generalization accuracy, but rather to enable individuals to audit how biased labels in the training data can affect their own outcomes.

A naive solution would be to enumerate all the alternate training datasets, retrain the model on each, and check whether the prediction changes. However, the number of such datasets grows combinatorially with the training size, and retraining

---

*Corresponding Author.

neural networks for each candidate is computationally expensive. Let $D$ be the original dataset with $n = |D|$, and let $m \leq n$ be the maximal number of examples whose labels may be biased. Then, the number of alternate datasets (and retrained models) is $\binom{n}{m}$, resulting in a worse-than-exponential blow-up.[1] Our goal is to avoid this explicit enumeration and instead efficiently find alternate datasets that can alter predictions.

To this end, we propose to analyze the impact of label bias on the training and inference of a neural network and derive heuristics from this analysis that can guide our search for an alternate dataset. As shown in Figure 1, our method takes the tuple $\langle D, \mathcal{L}, m, x \rangle$ and returns a new training dataset $D'$ as output, where $D$ is the original dataset, $\mathcal{L}$ the learning algorithm, $m$ the bound on the number of potentially biased labels, and $x$ the test input. First, our method follows the standard training and inference pipeline, where we train a network $f = \mathcal{L}(D)$ and compute a prediction $y = f(x)$. Second, the method iteratively constructs alternate datasets $D'$ based on our analysis techniques, retrains $f' = \mathcal{L}(D')$, and computes $y' = f'(x)$. If $y \neq y'$, $D'$ is returned; otherwise, the process repeats until a dataset yielding a different prediction is found.

We call this returned $D'$ a *counterfactual dataset* (CFD). If network training and inference are assumed fair, $D'$ can serve as a counterexample to that claim. The term *counterfactual* comes from mathematical logic, where the assertion *if $P$ then $Q$* can be rephrased as the counterfactual assertion *if $\neg Q$ then $\neg P$*, making evidence for $\neg Q$ a counterexample to $P$. Intuitively, a CFD demonstrates that the same learning algorithm $\mathcal{L}$ can produce a different outcome for $\mathbf{x}$ when trained on $D'$ instead of $D$ (i.e., $f'(\mathbf{x}) \neq f(\mathbf{x})$).

Our method generates $D'$ from $D$ using two complementary techniques. First, we use *linear regression as a surrogate model* to estimate the impact of label bias during the *training* stage. We exploit the piecewise linearity of ReLU networks and the closed-form solution of linear regression to rank which training labels have the greatest impact on the prediction of test input $x$. Second, we propose *neuron activation similarity*, which measures the distance between each training example in $D$ and the test input $x$ based on their neuron activation patterns. For ReLU networks, neurons are either *active* or *inactive* along the decision boundary of 0, while for other activation functions such as sigmoid and tanh, neuron activation can be approximated similarly via some threshold (i.e., *inactive* if below the threshold, or *active* if above). This similarity allows ranking examples to account for the network's *inference* behavior. Combining these two rankings, we

heuristically flip the labels of high-ranked training examples to generate alternate datasets $D'$.

We have implemented our method in PyTorch [29] using the Adam optimizer [30] and evaluated it on 7 datasets commonly used in fairness research. Our results show that our method generates counterfactual datasets efficiently: for smaller datasets, it quickly finds ground-truth CFDs, and for larger datasets, it outperforms baselines by producing CFDs across more test cases with negligible overhead. It also identifies training instances that better capture dataset label bias and remain faithful to the test input, and it succeeds even for inputs near and far from the decision boundary.

To summarize our contributions:

1. We propose a method that considers both the training and inference stages of deep neural networks to efficiently generate counterfactual datasets (CFDs), which provide evidence of potentially unfair predictions due to label bias.

2. We develop two complementary techniques to assess the impact of dataset label bias on a network's prediction for a given test input: linear regression as a surrogate model to analyze the training stage, and neuron activation similarity to analyze the inference stage.

3. We implement our method and demonstrate its effectiveness on diverse fairness datasets, showing how generated counterfactual datasets can explain changes in predictions.

## 2 Problem Setup

Let $D = (\mathbf{X}, \mathbf{y})$ be a training dataset with $n$ examples, and $\mathcal{L}$ be a learning algorithm that trains a neural network $f = \mathcal{L}(D)$. Some labels in $\mathbf{y}$ may be biased due to human subjectivity or historical prejudices. We denote by $m$ the maximal number of potentially biased labels. If $f$ is trained on such a dataset, its prediction for a given test input $\mathbf{x}$ may be biased, an example of dataset label bias [27, 28].

To formalize the scope of counterfactual dataset (CFD) generation, we define filtering rules $\phi$ for test inputs and $\psi$ for training examples. $\phi(\mathbf{x})$ specifies the test inputs eligible for CFD search, while $\psi(\mathbf{x}_i, y_i)$ identifies subset of training examples whose labels may be altered or are suspected of historical bias. These rules help ensure that the generated CFDs are meaningful. For example, $\phi$ can select test inputs that appear fair under *inference-only* methods (e.g., the network outputs the same label regardless of the protected attribute of $\mathbf{x}$), and $\psi$ can limit label changes to training examples similar to $\mathbf{x}$ in protected group and predicted outcome.

Given these rules, we say that $D'$, obtained by flipping up to $m \leq n$ labels of $(x_i, y_i) \in D$, is a

---

[1]If $n = 10{,}000$ and $m = 10$ (i.e., only 0.1% of labels may be biased), we already have $\binom{10{,}000}{10} = 2.74 \times 10^{33}$ datasets.

*counterfactual dataset* (CFD) if its retrained network $f' = \mathcal{L}(D')$ outputs a different prediction from the original network $f = \mathcal{L}(D)$ for the given $x$.

# 3   Related Work

**Fairness Testing and Verification:**   Prior work analyzes fairness of machine learning models from multiple angles. Fairness testing methods [6–11] are inference-stage-only approaches that systematically audit whether there exist specific inputs or groups that the model behaves less equitably. Fairness verification methods [12–16, 18, 31–33] also operate at the inference stage, providing formal guarantees of whether a model satisfies certain fairness constraints.

In contrast, our method looks at both the training and inference stages to serve as an auditing tool for individual test inputs, rather than aiming to achieve global fairness. Instead, it selectively changes a small number of training labels to generate counterfactual datasets (CFDs) that reveal potential unfair behavior for specific test cases. Conceptually, it can be viewed as a form of fairness testing applied at the training stage, producing CFDs to expose the impact of label bias.

**Training Data Manipulation and Correction:** Prior work has explored modifying training data to degrade or improve model behavior. Noisy label correction methods [34–37] aim to improve a model's accuracy by reducing the impact of mislabeled training data and treating noise as undesirable randomness. Data-poisoning techniques also manipulate training data, sometimes input features [38–40] and other times labels [41–46], but all share the goal of degrading overall model performance.

In contrast, our approach deliberately modifies a few labels in a targeted manner to generate CFDs that audit model behavior on specific test inputs. Rather than improving or degrading overall performance, our method tries to uncover potential fairness violations at the instance level.

**Explaining and Attributing Predictions:**   Independent of fairness research, there are several approaches to explain and attribute model outcomes. Counterfactual explanations [20–24] generate concrete examples with minimal changes to test inputs that would produce different model outputs. These can be understood as human-interpretable counterparts to adversarial examples [47–49], which also illustrate how small variations in input features can change decisions. Feature attribution techniques, such as LIME [50] and SHAP [51], assign importance scores to individual input features for a specific prediction to offer local explanations. While both are useful for interpretability, these methods

are concerned about changing or scoring specific features, rather than the impact of labels. Furthermore, they do not manipulate the training dataset to systematically audit model behavior, which is the focus of our work.

The methods most closely related to our approach are data attribution methods, which estimate the influence of each training sample on a model's prediction for a specific test sample. Among these, influence functions [52, 53] are a popular approach. Intuitively, they could be used to identify which training instances to prioritize in our CFD generation problem. However, unlike our method, they are not specifically designed to address the label bias problem. Furthermore, they are computationally expensive, particularly for nonlinear models such as multilayer perceptrons (MLPs) or residual networks (ResNets) [54]. We confirm this later, where our experiments show that various state-of-the-art influence functions [53, 55–57] are not only less effective than our method but also time-consuming, making them intractable for our setting of CFD generation for neural networks.

**Training Dataset Robustness and Fairness:** One line of work investigates the effect of including or leaving out a single training instance near the decision boundary on the newly trained model's predictions [58]. While related, our focus is distinct: we manipulate the *labels* of a subset of training examples to probe fairness under label bias, rather than analyzing the effect of instance removal.

Techniques more closely related to ours certify robustness and fairness under alternate training datasets [59–63]; in particular, [27, 28] focus on handling label-biased datasets. These methods verify whether the model's predictions for specific test inputs remain stable under defined perturbations of the training data. From this perspective, any perturbed training dataset that does cause a change in output can be interpreted as a counterfactual dataset, making these techniques conceptually related to our work. However, these approaches are designed for simpler models, such as decision trees, KNNs, and linear regression, and are not directly applicable to neural networks, due to the complexity of training and the high nonlinearity of their function representations.

**Summary:**   To the best of our knowledge, we are the first to efficiently generate counterfactual datasets (CFDs) for deep neural networks that uncover the impact of label bias on individual predictions. Unlike prior fairness works, our method audits both training and inference stages that is targeted towards a specific individual test case. Unlike dataset manipulation approaches, we do not seek to improve or degrade overall model performance. Unlike coun-

terfactual explanations or feature attribution, our focus is on training labels rather than input features at inference. Finally, while data attribution and dataset robustness techniques are conceptually related to our work, they are either computationally intractable or inapplicable for our setting.

# 4 Methodology

Our goal is to analyze the impact of label bias on neural network training and inference and to efficiently construct *counterfactual datasets* (CFDs) that provide interpretable evidence of unfair predictions. We begin with the key insight motivating our approach, then develop two complementary analysis techniques, and finally describe the top-level algorithm that leverages these analyses.

## 4.1 Key Insight

Let $\partial y$ denote the difference between predictions $y$ and $y'$ for a given test input, and let $\partial D$ denote the difference between two datasets $D$ and $D'$. Ideally, we want to characterize

$$\frac{\partial y}{\partial D} = \frac{\partial y}{\partial \mathbf{y}},$$

where $\partial \mathbf{y}$ is the difference between the original labels $\mathbf{y}$ in $D$ and the modified labels $\mathbf{y}'$ in $D'$. This derivative represents the impact of label bias on the prediction $y$.

Since closed-form analysis of stochastic gradient descent is intractable, we approximate this relationship via two heuristics:

1. **Linear regression surrogate:** provides a closed-form expression to measure the impact of label flips at the *training stage*.

2. **Neuron activation similarity:** measures closeness between test input and training examples at the *inference stage*.

Together, these heuristics guide our search for CFDs.

## 4.2 Piecewise Linear Approximations

Although neural networks are globally nonlinear, their inference-stage behavior for a specific input $\mathbf{x}$ may be approximated by a linear function. With ReLU activations, a neural network implements a piecewise linear function:

$$f(\mathbf{x}) = \begin{cases} \theta_1^\top \mathbf{x}, & \mathbf{x} \in D_1 \\ \vdots \\ \theta_p^\top \mathbf{x}, & \mathbf{x} \in D_p, \end{cases}$$

where $\{D_1, \ldots, D_p\}$ partition the input space and each $\theta_i^\top \mathbf{x}$ is linear for $\mathbf{x} \in D_i$. For a given $\mathbf{x}$, each

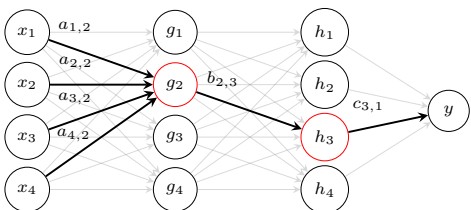

**Figure 2.** A neural network with ReLU activation for a given input $\mathbf{x}$. Active neurons are outlined in red.

hidden ReLU is either *active* ($\mathrm{ReLU}(z) = z$ if $z > 0$) or *inactive* ($\mathrm{ReLU}(z) = 0$ otherwise). Thus, the network $f$ can be restricted to the neurons activated by $\mathbf{x}$, yielding a local linear model.[2] As illustrated in Figure 2, this $\theta_i$ can be obtained as the product of active weights along the paths from input to output layer of the network.

Let $W^{(i,j)}$ denote the weight matrix from layer $i$ (with $n_i$ neurons) to layer $j$ (with $n_j$ neurons). Let $\mathbf{b}_i = [b_{i1}, \ldots, b_{in_i}]^\top$ and $\mathbf{b}_j = [b_{j1}, \ldots, b_{jn_j}]$ be the binary activation vectors of layers $i$ and $j$ when $\mathbf{x}$ passes through $f$, where $b_k = 1$ indicates that neuron $k$ is active and $b_k = 0$ otherwise. Then, the effective weight matrix is

$$\hat{W}^{(i,j)} = W^{(i,j)} \odot (\mathbf{b}_i \mathbf{1}_{1 \times n_j}) \odot (\mathbf{1}_{n_i \times 1} \mathbf{b}_j),$$

where $\odot$ denotes elementwise multiplication. In other words, only the rows (columns) corresponding to active neurons in layer $i$ ($j$) are retained. Multiplying across $l$ layers gives

$$\theta = \hat{W}^{(1,2)} \cdot \hat{W}^{(2,3)} \cdot \ldots \cdot \hat{W}^{(l-1,l)},$$

so that the network reduces to the linear model

$$y = f(\mathbf{x}) = \theta^\top \mathbf{x}.$$

This local linearization provides the basis for the techniques introduced in the following subsections.

## 4.3 Linear Regression Surrogate

Our first technique is motivated by prior work on analyzing dataset bias through linear regression [28, 60]. Unlike neural networks trained with stochastic gradient descent, linear regression admits a closed-form solution[3] for its optimal parameters:

$$\theta = (\mathbf{X}^\top \mathbf{X})^{-1} \mathbf{X}^\top \mathbf{y}.$$

Thus the prediction for $\mathbf{x}$ can be written as

$$y = \theta^\top \mathbf{x} = \mathbf{x}^\top (\mathbf{X}^\top \mathbf{X})^{-1} \mathbf{X}^\top \mathbf{y} = \mathbf{z}\mathbf{y} = \sum_{i=1}^{n} z_i y_i, \quad (1)$$

---

[2]For sigmoid or tanh activations, we can similarly approximate activations as binary by setting the threshold to some value (e.g., 0).

[3]Here, we show least-squares fit for readability, but in practice, we follow other works and implement ridge regression for stability: $\theta = (\mathbf{X}^\top \mathbf{X} - \lambda I)^{-1} \mathbf{X}^\top \mathbf{y}$.

where $\mathbf{z} = \mathbf{x}^\top (\mathbf{X}^\top \mathbf{X})^{-1} \mathbf{X}^\top$. Here, $z_i$ measures the contribution of training example $(\mathbf{x}_i, y_i)$ to the prediction $y$. Sorting training examples by $|z_i|$ yields a ranking of label influence.

Using this ranking, we can construct $\mathbf{y}'$ by sequentially flipping the top-ranked labels one at a time. For linear models, this procedure identifies the optimally flipped training label set under the bound $m$.[4] For nonlinear networks, the same reasoning provides a heuristic that remains effective due to piecewise linearity.

## 4.4 Neuron Activation Similarity

At inference, each input $\mathbf{x}$ induces a binary activation vector over hidden neurons, where each element indicates whether the corresponding neuron is active. We define the similarity between $\mathbf{x}$ and a training example $\mathbf{x}' \in \mathbf{X}$ as the inverse Hamming distance between their activation vectors:

$$\text{Sim}(\mathbf{x}, \mathbf{x}') = 1 - \frac{1}{d} \sum_{k=1}^{d} \mathbf{1}\{b_k(\mathbf{x}) \neq b_k(\mathbf{x}')\}, \quad (2)$$

where $d$ is the total number of hidden neurons and $b_k(\cdot) \in \{0, 1\}$ is the activation status of neuron $k$.

Intuitively, inputs with similar neuron activation patterns are likely to be governed by the same underlying network logic, as the activation status of neurons captures much of a feedforward network's behavior [64]. This perspective aligns with our discussion of piecewise linearity; we can expect that inputs with similar activation patterns lie in similar linear regions, so their $\theta$ in a linearized model are very close as well. We rank training examples by this similarity score to guide the search for labels that are most likely to affect the prediction of $\mathbf{x}$.[5]

## 4.5 Algorithm for CFD Generation

As described in Algorithm 1, our procedure for generating a counterfactual dataset (CFD) $D'$ takes as input: the dataset $D = (\mathbf{X}, \mathbf{y})$, a learning algorithm $\mathcal{L}$, a bias budget $m$, the test input $\mathbf{x}$, and filtering rules $\phi$ (for the test input) and $\psi$ (for training examples).

The procedure begins by training the baseline model $f = \mathcal{L}(D)$ and computing the original prediction $y = f(\mathbf{x})$, aborting if $\phi(\mathbf{x})$ is false. The predicate $\phi$ allows us to restrict attention to a subset of test inputs of interest. For instance, $\phi(\mathbf{x})$ could encode that the prediction $y$ is already deemed fair

---

**Algorithm 1** Generating counterfactual dataset.

1: **Input:** dataset $D = (\mathbf{X}, \mathbf{y})$, learning algorithm $\mathcal{L}$, bias budget $m$, test input $\mathbf{x}$, filtering rules $\phi$ and $\psi$
2: **Output:** counterfactual dataset $D' = (\mathbf{X}, \mathbf{y}')$
3: $f \leftarrow \mathcal{L}(D); y \leftarrow f(\mathbf{x})$ {original model & prediction}
4: **if** not $\phi(\mathbf{x})$ **then**
5:      **return** {$\mathbf{x}$ does not pass filtering by $phi$}
6: **end if**
7: $\mathbf{y}_L \leftarrow \text{LR\_SCORING}(\mathbf{X}, \mathbf{y}, \mathbf{x}, m)$ {Section 4.3}
8: $\mathbf{y}_A \leftarrow \text{ACTIV\_SCORING}(\mathbf{X}, \mathbf{x}, f)$ {Section 4.4}
9: $[y_1, \ldots, y_{n_\psi}] \leftarrow \text{COMBINE\_SCORING}(\mathbf{y}_L, \mathbf{y}_A, \psi)$
        {$n_\psi$ = size of training set after filtering by $\psi$}
10: $k \leftarrow 1$
11: **while** $k \leq m$ **do**
12:      $\mathbf{y}' \leftarrow$ label set where $y_1, \ldots, y_k$ in $\mathbf{y}$ are flipped
13:      $D' \leftarrow (\mathbf{X}, \mathbf{y}')$
14:      $f' \leftarrow \mathcal{L}(D'); y' \leftarrow f'(\mathbf{x})$ {new model & prediction}
15:      **if** $y \neq y'$ **then**
16:          **return** $D'$ {CFD solution found}
17:      **else**
18:          $k \leftarrow k + 1$ {flip more labels in next iteration}
19:      **end if**
20: **end while**
21: **return** {solution not found}

---

under standard inference-only checks; in this case, we only probe further on inputs that would otherwise pass fairness verification. As another example, $\phi$ may require that $\mathbf{x}$ belongs to a particular demographic group, enabling targeted analysis.

To guide label modifications, we score the training examples using two complementary analyses: $\mathbf{y}_L$ from the linear regression surrogate (Section 4.3) and $\mathbf{y}_A$ from neuron activation similarity (Section 4.4). The two scores are then combined and filtered according to $\psi$, producing a ranking of relevant training labels $[y_1, \ldots, y_{n_\psi}]$, where $y_1$ is expected to have the largest effect on the prediction for $\mathbf{x}$.

Iteratively flipping the top-$k$ labels in the ranking, for $k = 1$ to $m$, we create alternate datasets $D'$ and retrain networks $f' = \mathcal{L}(D')$ to obtain new predictions $y' = f'(\mathbf{x})$. The procedure terminates as soon as $y' \neq y$ and returns the current $D'$ as a CFD, or after reaching the budget $m$ without finding a valid CFD. By leveraging the rankings from our analyses, the method efficiently targets the label flips that are most likely to change the prediction.

## 4.6 Subroutines

**Linear Regression Scoring:** The LR\_SCORING subroutine sorts training examples by the magnitude of $z_i$ from Equation (1). High-$|z_i|$ examples have the largest effect on $\mathbf{x}$'s prediction.

**Activation Scoring:** The ACTIV\_SCORING subroutine sorts training examples by the magnitude of $\text{Sim}(\mathbf{x}, \mathbf{x}')$ from Equation (2). High similarity scores indicate that a training example shares a similar activation pattern with $\mathbf{x}$, placing it in a similar decision region in the network and making it more relevant for potential label flips.

---

[4]The resulting $\mathbf{y}'$ has a nice theoretical property for linear regressions; it is the optimal label set among the $\sum_{k=1}^{m} \binom{n}{k}$ possible sets. See [28] for proof.

[5]In practice, we apply a decay factor of $\frac{1}{2}$, assigning higher weights to neurons closer to the output; i.e., full weight given to the neurons in the $L - 1^{\text{th}}$ layer (1 if activation matches), half weight to the $L - 2^{\text{th}}$ layer (0.5 if match), quarter weight to the $L - 3^{\text{th}}$ layer (0.25 if match), and so on.

**Final Ranking:** We normalize and average $\mathbf{y}_L$ and $\mathbf{y}_A$ scores to obtain a combined ranking, balancing each training example's influence during the training stage with its proximity to the test input during the inference stage.

# 5 Experimental Evaluation

As we are the first, to the best of our knowledge, to consider the problem of generating counterfactual datasets for deep neural networks, there is no prior technique for direct comparison. Thus, we use exhaustive enumeration to set the ground truth for smaller datasets when feasible, and compare our method against six baselines:

- *Random Sampling*: A naive approach that selects training instances uniformly at random;

- $L_2$ *Distance*: A proximity-based approach that ranks training instances based on their Euclidean distance to the test point in the embedding feature space, computed using the `scikit-learn` library [65];

- *Explicit* [53], *Conjugate Gradients (CG)* [55], *LiSSA* [56], and *Arnoldi* [57]: Influence function methods that estimate, and subsequently rank, the influence of training instances on the test point via Hessian-based analysis, with *Explicit* computing the exact scores and the other three using various approximations. All four are available out-of-the-box through the `dattri` library [54].

We evaluate our method on seven widely used fairness datasets: Salary [66], Student [67], German [68], Compas [69], Default [70], Bank [71], and Adult [72]. Given our focus on fairness, we manually design the test and training filtering rules as described in Section 2 and assume up to 0.1% of training labels may be biased. In general, determining appropriate filtering rules ($\phi$ and $\psi$) and bias budget ($m$) to reflect real-world scenarios for auditing label bias requires domain expertise from the social sciences and is beyond the scope of this paper. Additional details of our experimental setup are provided in Appendix A.

## 5.1 Main Results

Our experiments were designed to answer the following research questions (RQs):

RQ1. Is our method *effective* in generating a CFD for a given test input?

RQ2. Is our method *efficient* in generating a CFD for a given test input?

**Table 1.** Comparison between our method and baselines across all datasets. $\#_\phi$: number of test inputs passing test input filter $\phi$ (max 200); $\boldsymbol{n}_\psi$: number of training examples passing training filter $\psi$; **GT**: number of ground truth CFDs found by exhaustive enumeration; **CFDs Found**: number of CFDs found by *Our Method* (in bold) / *Random Sampling* / $L_2$ *Distance*.

| Dataset | $\#_\phi$ | $\boldsymbol{n}_\psi$ | GT | CFDs Found |
|---|---|---|---|---|
| *Smaller Datasets* ($m = 1$, $t = 0.1 \times n_\psi$) | | | | |
| Salary | 10 | $[8, 9]$ | 3 | **3** / 1 / 3 |
| Student | 121 | $[23, 197]$ | 24 | **20** / 13 / 10 |
| German | 182 | $[48, 267]$ | 38 | **38** / 17 / 15 |
| *Larger Datasets* ($m = 0.1\% \times n$, $t = 10m$) | | | | |
| Compas | 200 | $[225, 887]$ | – | **27** / 10 / 5 |
| Default | 200 | $[612, 8219]$ | – | **18** / 5 / 8 |
| Bank | 200 | $[512, 8951]$ | – | **24** / 9 / 11 |
| Adult | 200 | $[532, 11505]$ | – | **44** / 15 / 15 |

RQ3. Does our method generate a CFD $D'$ for a given test input that is *meaningful*?

RQ4. Can our method generate CFDs for test inputs near or far from the *decision boundary*?

### 5.1.1 Results for RQ1

To answer RQ1 about the *effectiveness* of our method, we compare its success rate at finding valid counterfactual dataset (CFDs) against the baselines. Table 1 summarizes our main findings on both smaller and larger datasets, reporting the number of test inputs passing $\phi$, the number of training examples passing $\psi$,[6] the number of ground-truth CFDs where exhaustive enumeration is feasible, and the number of CFDs successfully found by each method. Across the datasets, we evaluate our methods on over 1100 test inputs (the sum of Column 2), which highlights the extensive scope of our evaluation.

We omit the results of the four influence function methods because all of them were computationally intractable. For instance, on the Student dataset, *Explicit* requires roughly 2 hours per test case, *CG* 3 hours, *LiSSA* 70 hours, and *Arnoldi* just under 1 hour, even before accounting for retraining. Multiplying these costs by the number of test cases to evaluate ($\#_\phi$) quickly becomes infeasible, especially given that these runtimes arise on a relatively small dataset like Student. This computational intractability for nonlinear models such as neural networks aligns with prior findings reported in [54].

**Smaller Datasets** For the smaller datasets where only a single label flip is allowed ($m = 1$), we set the maximum number of iterations $t$ as $0.1 \times n_\psi$. As a result, $t = 1$ for Salary and $t = [2, 19]$ and $t = [4, 26]$ for Student and German respectively.

---

[6]$n_\psi$ is represented as an interval because the number of training examples that pass $\psi$ vary across the test cases.

We slightly modify our algorithm so that at iteration $t$, the ranking-based methods (i.e., our method and the $L_2$ distance baseline) flip the label of the top $t$-th ranked training example rather than increasing $k$. In contrast, random sampling may select any of the $n_\psi$ candidates at each iteration.

Exhaustive enumeration is feasible here since $m$ and $n_\psi$ are small. Looking at Columns 4-5, we observe that our method finds a CFD for nearly all individuals with ground-truth CFDs, missing only 4 cases in Student (20/24). Furthermore, our approach substantially outperforms both random sampling and $L_2$ distance. For example, in the German dataset our method identifies all existing CFDs, while both baselines succeed on less than half the test cases (17 for random sampling and 15 for $L_2$ distance, out of 38).

**Larger Datasets** As exhaustive enumeration is infeasible for the larger datasets, Column 4 is omitted, and we evaluate all the methods under $t = 10m$ iterations. In other words, there are 10 attempts for each $k \leq m$, where $k$ is the number of labels flipped.

We again slightly modify our algorithm so that at the first attempt for a given $k$, the ranking-based methods (our method and $L_2$ distance) flip the top-$k$ labels. In the remaining 9 attempts, it samples among the top-$ak$ labels (for the smallest $a$ with $\binom{ak}{k} > 10$) to ensure sufficient diversity across attempts.[7] For a fair comparison, random sampling is also given 10 independent attempts per $k$.

Under this setting, our method uncovers CFDs for 9–22% of test inputs, compared to only 2.5–7.5% for random sampling and $L_2$ distance. For reference, the ground truth on smaller datasets shows CFDs for 16–30% of test inputs, suggesting that our results for larger datasets represent conservative lower bounds.

> **Answer to RQ1:** Our results demonstrate that our analysis-guided search is significantly more effective at discovering CFDs than the baselines.

### 5.1.2 Results for RQ2

To answer RQ2 on the computational *efficiency* of our method, we report and compare runtimes (in seconds) across methods in Table 2. Column 2 reports the average runtime per test case. Column 3 captures the average runtime per training iteration, enabling a fair comparison across methods with different iteration counts. Column 4 isolates the non-training overhead, highlighting the computational efficiency of each method independent of retraining.

Per-test and per-iteration times (Columns 2-3) are nearly identical across methods (within 0.3s

---

[7] The condition $\binom{ak}{k} > 10$ guarantees that there are at least 10 distinct ways to choose $k$ from the top-$ak$ labels. Without this, sampling could collapse to only a handful of options, limiting diversity and making the comparison unfair.

**Table 2.** Runtime comparison between our method and baselines across all datasets, in order of: *Our Method / Random Sampling / $L_2$ Distance.* **Per-test Time**: average runtime per test case; **Per-iter Time**: average runtime per training iteration per test case; **Overhead**: average non-training overhead runtime per test case.

| Dataset | Per-test Time (s) | Per-iter Time (s) | Overhead (s) |
|---------|-------------------|-------------------|--------------|
| *Smaller Datasets* | | | |
| Salary | 0.35 / 0.31 / 0.32 | 0.35 / 0.31 / 0.32 | 0.04 / 0.00 / 0.01 |
| Student | 4.19 / 4.35 / 4.52 | 0.32 / 0.31 / 0.32 | 0.07 / 0.02 / 0.03 |
| German | 2.33 / 2.56 / 2.62 | 0.16 / 0.15 / 0.16 | 0.08 / 0.02 / 0.04 |
| *Larger Datasets* | | | |
| Compas | 14.83 / 17.03 / 17.18 | 0.40 / 0.44 / 0.44 | 0.28 / 0.12 / 0.24 |
| Default | 133.82 / 134.34 / 122.55 | 0.80 / 0.76 / 0.73 | 2.91 / 1.52 / 4.32 |
| Bank | 123.53 / 143.85 / 142.38 | 0.73 / 0.79 / 0.82 | 3.22 / 1.72 / 5.04 |
| Adult | 195.81 / 206.90 / 206.32 | 0.96 / 0.79 / 1.00 | 5.73 / 3.26 / 9.35 |

and 0.04s, respectively, on smaller datasets). On larger datasets, our runtimes remain on par with the baselines and can even be lower than random sampling (e.g., *Compas* and *Bank*). This advantage grows with dataset size, where longer retraining times and higher iteration budgets ($t$) make early CFD discovery especially beneficial. By succeeding on more test cases, our method often terminates earlier, reducing total runtime.

Our method does introduce some overhead (Column 4) relative to random sampling. However, this overhead remains small (within 0.1s on small datasets and 1-6s on large datasets) and becomes comparable to, or even lower than, $L_2$ distance as dataset size increases. Since the ranking analysis in Algorithm 1 is executed only once before retraining begins, its one-time upfront cost does not compound across iterations. Overall, these results suggest that our ranking analysis is not a bottleneck and that retraining remains the dominant runtime component across all methods and datasets.

> **Answer to RQ2:** Our method matches the baselines in per-test and per-iteration runtime. The minor overhead introduced is minor and remains negligible relative to model retraining.

### 5.1.3 Results for RQ3

To answer RQ3 on the *meaningfulness* of the generated CFDs, we compare the training examples selected by our method and the baselines against their test inputs. While $L_2$-distance ranking seems a reasonable heuristic for prioritizing nearby examples in the embedding space, our results show otherwise.

Representative results are shown in Figure B.1. The green line represents the test input; blue indicates the training examples selected by our method; orange and pink correspond to random sampling and $L_2$ distance, respectively; and red highlights any additional viable CFDs discovered through exhaustive enumeration. The $x$-axis presents the input attributes, and the $y$-axis presents their values.

In Figure B.1(a), random sampling fails to find any CFD, while our method (as well as $L_2$ distance) correctly selects the top-ranked training example, which matches all categorical features and is numerically closer than the alternative found by exhaustive enumeration. Figure B.1(b) shows that both methods find CFDs, but our method selects an example that is substantially closer on the numerical attributes. The $L_2$ baseline selects an example that matches well on the categorical features but significantly diverges on the numerical features, identifying a less representative example.

In Figure B.1(c), all methods find a valid CFD, but our method's example aligns more closely with the test input across numerical and categorical features. Specifically, all categorical features match with the test input, and the only differences are in *age* and *hours-per-week*. Even for these two, our example is as close as, if not closer to, the test input than the ones chosen by the baselines. Figure B.1(d) shows random sampling's chosen training example, whereas our method selects two examples within the budget that together form a CFD and more faithfully represent the test input. One of our chosen examples is identical to the test input (in light blue) and and the other differs only in having 2 fewer *priors_count* (in blue). In contrast, the random sampling example differs in *age_cat* and has 0 *priors_count*, making it less representative of the test input.

> **Answer to RQ3:** Our method generates CFDs that are not only valid but also more meaningful, selecting training examples that are closer and more representative of the test input than alternatives chosen by the baselines.

### 5.1.4 Results for RQ4

To answer RQ4 on how our method performs relative to a test input's distance from the *decision boundary*, we analyze the distribution of original network logits $y$ (i.e., values before the sigmoid for binary labels) for test inputs with CFDs found by our method and the baselines. As shown in Figure B.2, boxes represent the first, second, and third quartiles, and whiskers show the minimum and maximum values.

Test inputs with $y$ values near 0 lie close to the decision boundary, whereas those with larger absolute $y$ values are farther from it. Intuitively, generating a CFD is easier for inputs near the decision boundary. Figure B.2 demonstrates that our method can find CFDs even for *harder* test inputs, those farther from the boundary more reliably than random sampling and $L_2$ distance. This reinforces that our method not only is more *effective*, *efficient*, and produces more *meaningful* CFDs, but also *succeeds* on test inputs that are challenging for baseline methods.

> **Answer to RQ4:** Our method reliably generates CFDs even for test inputs that are far from the decision boundary, outperforming random sampling on these more challenging cases.

## 5.2 Additional Results

We refer the readers to Appendix E for a detailed ablation study, which sheds light on how different components of our method contribute to performance. Our results show that incorporating both training (Section 4.3) and inference (Section 4.4) allows our method to more effectively rank influential training labels than either ablation alone.

Looking at our results closely, Appendix D highlights that our method not only finds a large number of CFDs overall, but also captures nearly all CFDs discovered by baseline approaches while identifying many additional cases unique to our method, which underscores the comprehensive coverage of our approach. Complementing this, Appendix C shows that our method often succeeds in a single iteration $t = 1$, unlike the baselines. This *one-shot success* demonstrates the effectiveness of our rankings in prioritizing impactful labels.

Importantly, changing a single training label can be sufficient to flip an individual's prediction, highlighting the sensitivity of model decisions to even minimal bias in the training labels. This raises a key concern for fairness-sensitive applications: small labeling biases or inaccuracies before training time can propagate into disproportionately large consequences at inference time. Beyond serving as a way to measure efficiency, the prevalence of one-shot CFDs illustrates how fragile decision boundaries can be, and why methods that can reliably surface such cases are critical for auditing model fairness.

## 6 Conclusion

We present a method for analyzing how biased training labels can affect neural network predictions by generating counterfactual datasets (CFDs). Our approach targets specific training labels to audit the network's fairness on *individual test inputs*. To avoid exhaustively enumeration and repeated retraining, it leverages a linear regression surrogate and neuron activation similarity to rank training examples for CFD generation. By considering both the training stage (via linear regression) and the inference stage (via activation similarity), our method prioritizes the training examples most likely to influence a test input's prediction. Experimental evaluation on a diverse set of fairness datasets demonstrates that our method is both effective and efficient, generating meaningful CFDs with minimal modifications to the original data.

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

# A  Experimental Details

**Table A.1.** Statistics of the fairness research datasets and training setup used in our experiments. **PA** = protected attribute; **# Tr, Va, Te** = numbers of training, validation, and test instances; $m$ = bound; **Input** = number of attributes/neurons in the input layer; **Arch** = network architecture (number of hidden layers × neurons).

| Dataset | PA | # Tr, Va, Te | $m$ | Input | Arch |
|---|---|---|---|---|---|
| Salary [66] | Sex | 30, 11, 11 | 1 | 5 | $2 \times 4$ |
| Student [67] | Sex | 389, 130, 130 | 1 | 36 | $2 \times 32$ |
| German [68] | Sex | 600, 200, 200 | 1 | 36 | $2 \times 32$ |
| Compas [69] | Race | 3702, 1235, 1235 | 4 | 13 | $2 \times 16$ |
| Default [70] | Sex | 18000, 6000, 6000 | 18 | 27 | $2 \times 32$ |
| Bank [71] | Marital | 18292, 6098, 6098 | 19 | 35 | $2 \times 32$ |
| Adult [72] | Sex | 27132, 9045, 9045 | 28 | 28 | $2 \times 32$ |

We have implemented our method as a Python tool and conducted all experiments on a computer with Intel Xeon W-2245 8-core CPU, NVIDIA RTX A5000 GPU, and 128GB RAM, running the Ubuntu 20.04 operating system.

As summarized in Table A.1, we use seven datasets widely used in fairness research for evaluation.[8] The top half shows three smaller datasets (Salary, Student, and German), while the bottom half shows four larger datasets (Compas, Default, Bank, and Adult). Each dataset is split into 60% training, 20% validation, and 20% test sets, as shown in Column 3. We set the bias budget $m$ is set to 0.1% of the training set size, as shown in Column 4, assuming that up to 0.1% of the training data may be biased. We preprocess the datasets by removing empty values, normalizing numerical features, and embedding categorical features, with the resulting input sizes shown in Column 5.

We train binary classifiers using PyTorch [29] with the Adam optimizer [30], fixing random seeds for determinism, which lets us isolate the effects of CFD generation under a fixed learning algorithm $\mathcal{L}$. We also allow early stopping for best validation loss, with a patience value of 10, learning rate of 0.005, and the number of epochs set to 100 (200) for smaller (larger) datasets. For network architecture, there are two hidden layers with layer sizes chosen relative to input dimensionality, as shown in Column 6.

# B  Experimental Figures

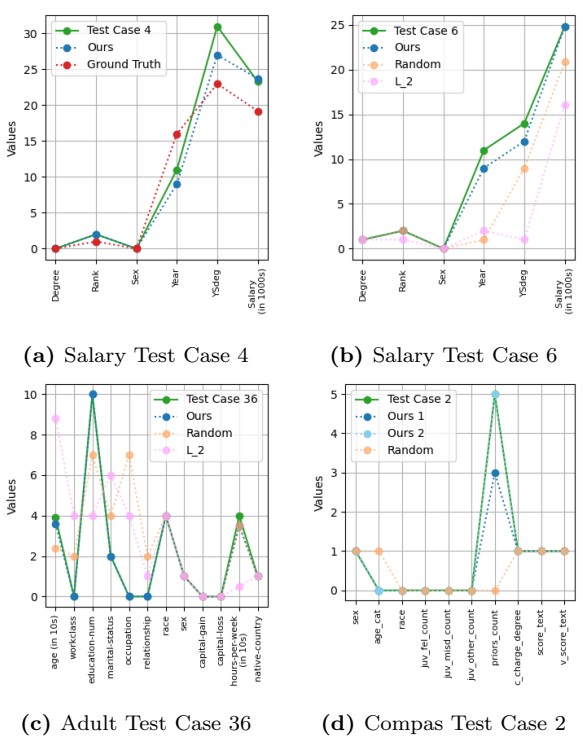

**(a)** Salary Test Case 4   **(b)** Salary Test Case 6

**(c)** Adult Test Case 36   **(d)** Compas Test Case 2

**Figure B.1.** Comparison of training examples selected by our method and baseline approaches for CFD generation. Each plot shows feature values of the test input alongside the chosen training examples.

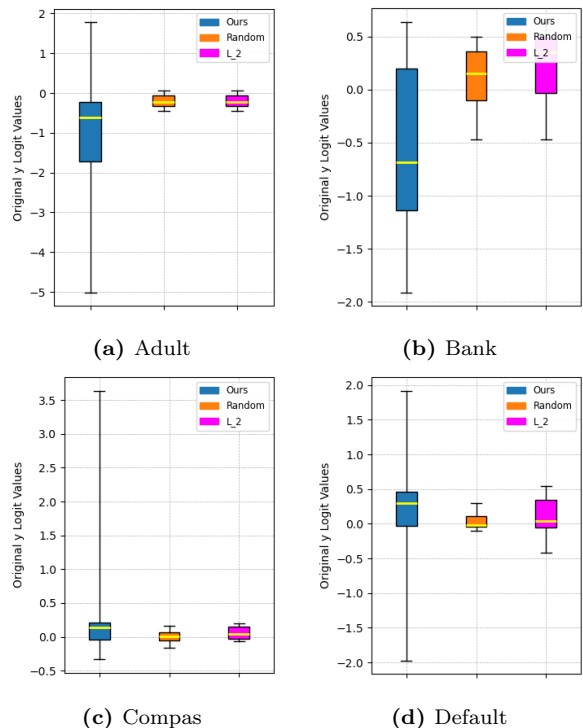

**(a)** Adult   **(b)** Bank

**(c)** Compas   **(d)** Default

**Figure B.2.** Comparison of original $y$ logit values for successful cases for various datasets. Boxplots show quartiles (boxes) and min/max ranges (whiskers). Colors denote methods: blue for *Our Method*, orange for *Random Sampling*, and magenta for $L_2$ *distance*.

---

[8]Refer to [4] for a survey of fairness datasets.

# C One-Shot CFD Comparison

# D Overlap of CFDs

**Table C.1.** One-shot comparison ($t = 1$) between our method and baselines across datasets. $\#_\phi$: number of test inputs passing the filter $\phi$; **CFDs Found**: number of CFDs found by *Our Method* in bold / *Random Sampling* / $L_2$ *Distance*; **One-Shot CFDs**: subset of *CFDs Found* that require exactly $t = 1$ iteration.

| Dataset | $\#_\phi$ | CFDs Found | One-Shot CFDs |
|---------|-----------|------------|----------------|
| *Smaller Datasets* | | | |
| Salary | 10 | **3** / 1 / 3 | **3** / 1 / 3 |
| Student | 121 | **20** / 13 / 10 | **9** / 1 / 3 |
| German | 182 | **38** / 17 / 15 | **20** / 4 / 0 |
| *Larger Datasets* | | | |
| Compas | 200 | **27** / 10 / 5 | **3** / 0 / 0 |
| Default | 200 | **18** / 5 / 8 | **1** / 1 / 1 |
| Bank | 200 | **24** / 9 / 11 | **4** / 3 / 2 |
| Adult | 200 | **44** / 15 / 15 | **7** / 3 / 4 |

Table C.1 provides the per-dataset statistics on one-shot success (i.e., the number of test inputs for which a CFD is found in the very first iteration) of our method against the baselines. Our method consistently identifies more CFDs overall and does so in fewer iterations than the baselines, demonstrating both efficiency and effectiveness. Notably, the prevalence of one-shot CFDs illustrates that even minimal changes to a single training label can flip model predictions, highlighting the sensitivity of decision boundaries. This reinforces the need for methods that can reliably surface such cases, as they are crucial for auditing model fairness and ensuring robust model behavior.

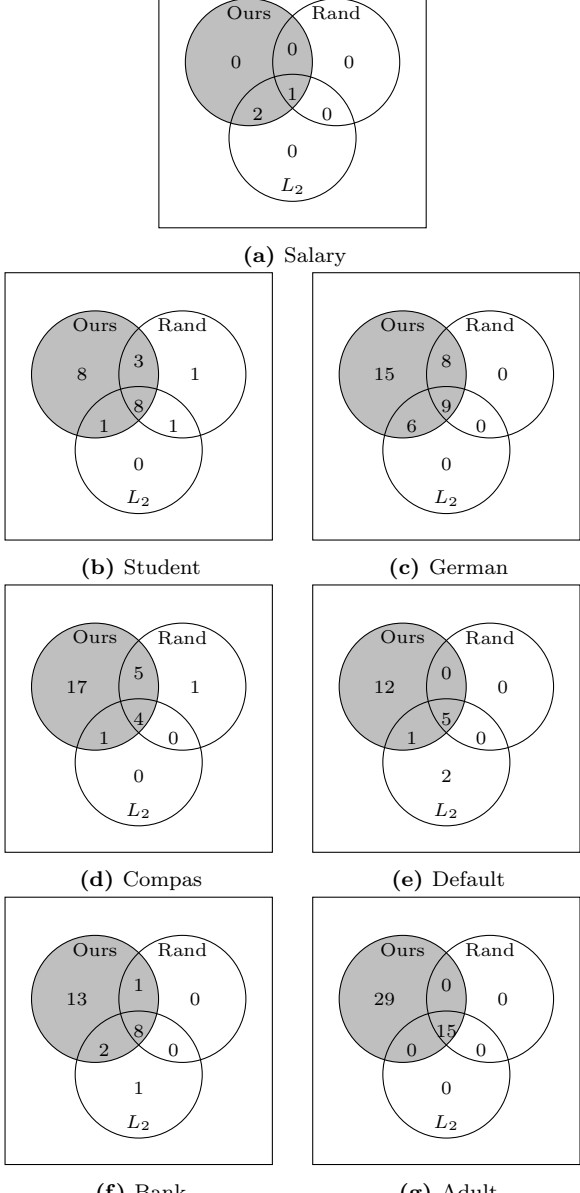

**(a)** Salary

**(b)** Student     **(c)** German

**(d)** Compas     **(e)** Default

**(f)** Bank     **(g)** Adult

**Figure D.1.** Venn diagrams showing overlap of CFDs found by Our Method (filled), Random Sampling, and $L_2$ Distance across datasets. Numbers inside each region indicate the total CFDs found, as reported in Table 1.

As shown in Figure D.1, our method identifies many CFDs that it uniquely finds; in contrast, CFDs found exclusively by Random Sampling or $L_2$ distance are extremely rare, typically only one or two instances. This indicates that almost all CFDs discovered by these baseline methods are also captured by our method. This pattern is consistent across all datasets, demonstrating that our method not only finds more CFDs overall, but also covers the solution space of other approaches.

# E    Ablation Study

**Table E.1.** Ablation study on smaller datasets ($m = 1$). $\#_\phi$: number of test inputs passing the filter $\phi$; **CFDs Found**: number of CFDs found by *Our Method* in bold / $\mathbf{y}_L$-*score only* / $\mathbf{y}_A$-*score only*; **One-Shot CFDs**: subset of *CFDs Found* that require exactly $t = 1$ iteration.

| Dataset | $\#_\phi$ | CFDs Found | One-Shot CFDs |
|---|---|---|---|
| Salary | 10 | **3** / 3 / 1 | **3** / 3 / 1 |
| Student | 121 | **20** / 18 / 15 | **9** / 8 / 5 |
| German | 182 | **38** / 38 / 34 | **20** / 20 / 18 |

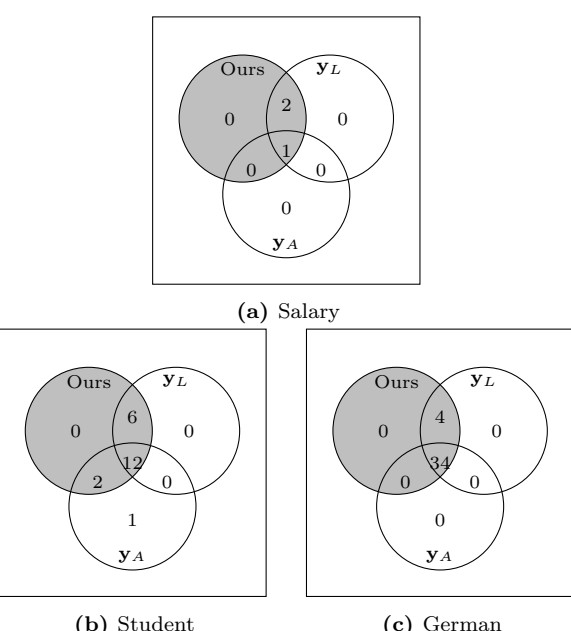

**(a)** Salary

**(b)** Student          **(c)** German

**Figure E.1.** Venn diagrams showing overlap of CFDs found by Our Method (filled), $\mathbf{y}_L$ only, and $\mathbf{y}_A$ only across smaller datasets. Numbers inside each region indicate the total CFDs found, as reported in Table E.1.

Table E.1 shows the results of an ablation study on the smaller datasets. Column 2 shows that combining our two proposed techniques consistently finds more CFDs than using either the linear regression surrogate ($\mathbf{y}_L$-score) or neuron activation similarity ($\mathbf{y}_A$-score) alone. Column 3 further confirms this advantage in terms of one-shot success (i.e., the number of test inputs for which a CFD is found in the very first iteration).

The overlap analysis in Figure E.1 further illustrates this trend. Our combined method misses only a single test case with a CFD (in Student), whereas $\mathbf{y}_L$ misses 3 (also in Student) and $\mathbf{y}_A$ misses 12 in total across datasets. This demonstrates that integrating both components provides broader and more reliable coverage.

For a detailed comparison, Figure E.2 presents a heatmap-style bar chart showing the number of test

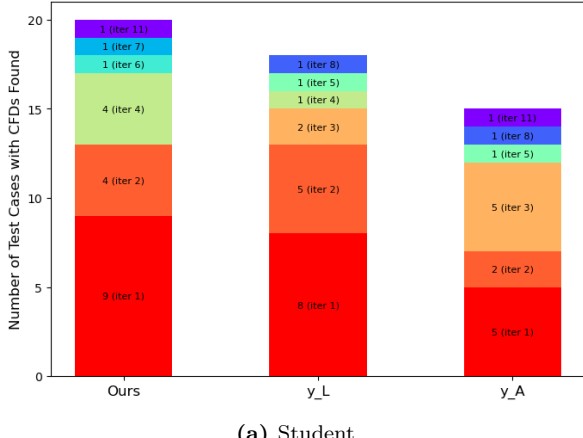

**(a)** Student

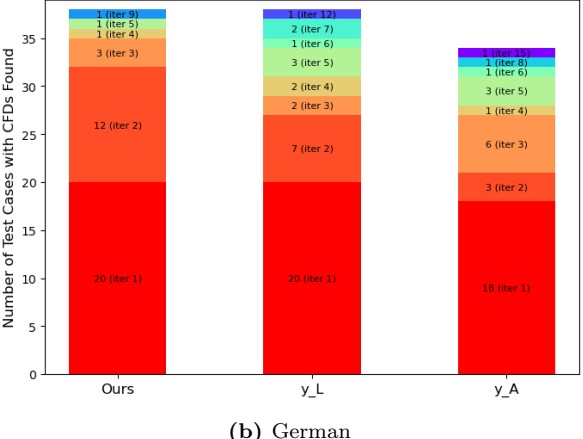

**(b)** German

**Figure E.2.** Number of test cases with CFDs found by the three variants of our method for Student and German datasets. Red (purple) colors mean that the CFDs were found at earlier (later) $t$-th iteration.

inputs with CFDs found (one per test input) by the three ablation methods. The total height of each stacked bar corresponds to the values in Column 2 of Table E.1, while the color shading reflects the training iteration $t$ (red for earlier and purple for later iterations). Since these smaller datasets have $m = 1$, flipping the $t$-th ranked training label at iteration $t$ directly indicates whether the method is prioritizing impactful labels in its ranking. Thus, a larger number of CFDs identified in earlier iterations signals greater efficiency.

As expected, our method (leftmost bar), which combines the two scores, consistently finds the most CFDs across $t$ values. For German dataset, for example, both our method and $\mathbf{y}_L$-score only find 20 CFDs in $t = 1$, but by $t = 2$, our method pulls ahead and identifies 12 new test inputs with CFDs compared to 7. This illustrates that by jointly considering training and inference, our method more effectively identifies influential training labels than the ablation variants.

