# OpenReview forum: "Analyzing Fairness of Neural Network Prediction via Counterfactual Dataset Generation"
_NLDL.org/2026/Conference — NLDL 2026 Spotlight_

### Official Review · Reviewer_L1sa · 2025-10-01
**Worthwile method but weak evaluation and connection to literature**

**Rating:** 4
**Confidence:** 3
**Final Rating:** 4
**Final Confidence:** 3

**Summary:**

The authors propose a method to create counterfactual datasets (CFDs), i.e. variants of the original dataset where labels are flipped in a systematic way, s.t. when the model is retrained on the CFD, the prediction for a given test datapoint changes. This allows to audit which training datapoints' labels most influence a particular test prediction. The framing is, that this may help find label bias.
Two methods are devised to estimate the impact of training label flips, in order to generate CFDs: (1) the importance of the training point to a linear surrogate model, and (2) the activation similarity to the test point. The resulting scores are combined to determine flipping order.
Evaluations on 7 tabular datasets show that small prediction-flipping CFDs can be found for many test samples, with computational cost comparable to naive baseline methods.

**Strengths:**

- The paper is clearly structured. The separation into RQs makes the experimental section easy to follow.
- The method itself appears novel, and the motivation is convincing.
- There is no obvious flaw in terms of correctness.
- Overall, the approach and experiments seem reproducible, although I would encourage the authors to make the code available upon publication.
- The proposed method often finds counterfactual datasets in one or a few iterations -- which is a reassuring observation, given the computational demands of repeated retraining.

**Weaknesses:**

- The paper lacks connections to relevant and established explainability methods. (1) influence functions, which trace a prediction to the training datapoints, appear to be crucial related work that needs to be discussed. (2) the linear surrogate model approach is the basis for many established attribution methods (saliency maps, gradient-times-input, layer-wise relevance propagation, guided backpropagation, …), some of which may be proper to reference as well.
- In Fig. 3b, it is surprising that the training sample selected by the proposed methods is closer to the test case than the one found by L2 distance. (1) It should be stated more clearly that L2 is calculated in the embedding space, when introducing the baseline, and (2) this evaluation (framing sample-similarity as a positive solution property) opens the question of why not ranking the samples to flip based on their input-space distance instead. Perhaps the authors could comment on the latter.
- While the framing is related to fairness, none of the experiments actually show that the found influential datapoints/labels are actually violating fairness or introducing bias, and no qualitative insight was obtained with the proposed method. (Besides the fact that, in some cases, mislabeling a single training sample suffices to change the prediction for a particular test sample.)
- [Minor] The proposed method does not appear to be very scalable, due to repeated retraining.
- [Minor] It is unclear how well the obtained linear surrogate model represents the learned network.
- [Minor] It should be better described how exactly the decay factor is implemented in the neuron similarity metric. Perhaps the authors could comment on this, as well as the motivation for doing so.

**Final Justification:**

The presentation in the paper is clear, and the method appears novel and correct. Some of the experimental results point to promising efficiency of the method in terms of the number of iterations (although retraining cost is still high).
While additional literature urgently needs to be addressed, the authors promised to do so. And while I am still not convinced by the connection to fairness, this is only a matter of framing, and the authors promised to clarify this.

**Justification:**

The major weaknesses of the paper are the lack of connection to relevant & well-established literature (especially influence functions) and the lack of insights/utility obtained using the method. In a setting designed for the particular method, naïve baselines are outperformed, but this gives little indication of the method's actual utility for fairness/bias evaluation.

Nonetheless, the approach appears novel enough and is well-described, giving ample opportunity for improvements and more thorough evaluations in future work. The experimental setup makes sense, and the results clearly support the method. Overall, CFDs are a worthwhile angle on bias evaluation.

---

> ### Author Rebuttal · Authors · 2025-10-22
>
> We thank Reviewer L1sa for their thoughtful response.   Here are our answers to the questions.
>
> * In our initial evaluation, we experimented with influence functions using the Python package 'dattri' [Deng et al., NeurIPS 2024], which implements several popular frameworks, including Explicit Computation, Conjugate Gradients, LiSSA, and Arnoldi.  While these methods can, in principle, identify training points relevant to specific test outcomes, we found them highly impractical for deep learning applications, a finding consistent with the evaluation reported in the 'dattri' paper.  In practice, these approaches required several hours per test instance merely to determine which training points to relabel, making them infeasible for our comparison scope.  We are happy to include additional details about this during the revision process upon acceptance.
> * For the L2 distance baseline, we do mention that this is in the embedding feature space in the beginning of Section 5 (line 397), but it might currently be getting lost in the details.  We will highlight this further during the revision process upon acceptance.  We chose to use the embedding-space distance rather than input-space distance because our goal was to evaluate if the model's "learned representation" could more effectively guide the search process.  In this sense, the embedding space provides a more meaningful notion of similarity, as it reflects how the model internally distinguishes between inputs.  Moreover, our training filter \psi already restricts the candidate test inputs to those relevant to the target point in terms of influential input-space features (e.g., protected attribute) and can be customized further.  Thus, using embedding-space distance served us as a complementary approach, and based on our results, we report that it was not effective in the context of CFD.
> * In fairness literature, dataset label bias is defined as examining whether flipping certain training labels and retraining the model leads to different outcomes for a given test sample [25, Li et al., CAV 2023; 26, Meyer et al., FAccT 2023].  Under this definition, the training data labels we identify through our CFD discovery process do violate this fairness property.  This notion may differ from other various definitions of fairness or bias, and we acknowledge this distinction and frame our work in the context of dataset label bias explicitly in the Introduction.  We can highlight this positioning even more during the revision process upon acceptance.  * Regarding the note about how "mislabeling a single training sample suffices to change the prediction for a particular test sample", we view this as a key insight rather than a limitation.  In fact, our finding aligns with the Leave-One-Out Fairness framework [Black et al., FAccT 2021], which investigates how removing a single training instance entirely can alter model predictions.  Similarly, our result shows that even a single mislabeled training point can have a significant impact on a test prediction, underscoring the sensitivity of model behavior to label bias and the importance of identifying such influential data points.  We can emphasize this insight during the revision process upon acceptance as well.
> * We agree with the reviewer that model retraining is computationally expensive, which may make our proposed approach appear impractical.  However, this computational burden is precisely the motivation for our work: since each retraining is so costly, it is crucial to identify the right data points to flip in order to obtain a viable CFD.  As we stated in the Problem Formulation, the overhead of model retraining is inherent to the problem definition itself, not to our proposed method.  Our heuristics-based approach, which runs in only a few seconds within the overall pipeline, aims to minimize the number of retraining operations required by efficiently guiding the search toward relevant data points.  Therefore, the retraining cost should not be viewed as a limitation of our approach's scalability, but rather as an aspect of the problem setup.
> * For this work, we decided to use linear surrogate models, as ReLU networks are known to be piecewise-linear functions across their input domains.  We viewed the linear surrogate model as a coarse approximation of the behavior of the ReLU network.  We empirically noticed that the results carried over well for non-piecewise linear activations, such as sigmoid.  We agree with the reviewer that there is definitely more to explore here in terms of understanding how well surrogate models capture the intended behavior of the more complicated system, and we leave this exploration to future work.
> * As noted in the footnote on pg. 4, we apply a decay factor of ½ across layers, assigning higher weights to neurons closer to the output.  In practice, this means that neurons in the L-1 layer receive the full weight (1 if activation matches), layer L-2 receive half the weight (½ if activation matches), layer L-3 receive quarter the weight (¼ if activation matches), and so on.  The motivation for this design is to emphasize neuron similarity in the outer layers, as activations closer to the output are more indicative of the model's final decision.  In our initial evaluation, removing the decay factor yielded comparable but slightly less favorable results, supporting the inclusion of this decay factor.  We're happy to clarify this during the revision process upon acceptance.

---

### Official Review · Reviewer_fMrK · 2025-10-10
**A decent paper introduces a new way to check if a machine learning model's decision is fair.**

**Rating:** 4
**Confidence:** 3

**Summary:**

To check if a machine learning model's decision is fair, Instead of asking, "What if we change the person's details (e.g., race, gender) to see if the prediction changes?", the paper asks, "What is the smallest change we can make to the original training data that would flip the prediction for this person?".
The method finds a few training examples whose labels might be biased, flips them, retrains the model, and shows that the outcome for a specific person changes.

**Strengths:**

The primary strength of this paper lies in its novel and effective method for assessing model fairness. The authors introduce a compelling paradigm shift, moving the focus of fairness analysis from inference-stage behavior to the training data itself. Their core proposa`` to reveal fairness issues by efficiently finding the closest alternative training dataset that alters a prediction'' is both intuitive and powerful.

This shift is cleverly learned from label correction.

Given the originality of the shift claimed by authors (sorry I can not identify if it does original) formulation, it is conceivable that this work, perhaps with a expansion of its experimental scope, would meet the high standards of machine learning conferences like ICLR or ICML, or more theoretical expansion for JMLR .

**Weaknesses:**

The primary limitation of the work is that the validation, while compelling, feels more like a proof-of-concept than a scalable solution. The approach hinges on model retraining, which is computationally expensive even for a single instance. This cost becomes a significant barrier when one considers the requirements of a truly systematic fairness audit. Such an audit would not be a one-off check; it would require iteratively applying the method across a comprehensive set of individuals, particularly those in sensitive subgroups, and potentially evaluating fairness against multiple definitions or with respect to various protected attributes.
The cumulative cost of such a comprehensive evaluation would likely be prohibitive even for the simple feedforward networks used in the paper. This scalability challenge is, of course, magnified exponentially for the large-scale foundation models where fairness concerns are most acute, rendering the current approach impractical for many of the most critical real-world applications.

**Justification:**

Unfortunately, I am a core ml guy, not an expert in fairness.
I have very little background knowledge of the field.
Without considering this background information, judging by the information in the paper, I believe the contribution of this paper is clearly meets the acceptance criteria.

---

> ### Author Rebuttal · Authors · 2025-10-22
>
> We thank Reviewer fMrK for their thoughtful response.
>
> We agree with the reviewer that model retraining is computationally expensive even for a single instance, especially with today's large-scale foundation models, which may make our proposed approach appear impractical.  However, this computational burden is precisely the motivation for our work: since each retraining is so costly, it is crucial to identify the right data points to flip in order to obtain a viable CFD.  Our heuristics-based method avoids such overhead by efficiently guiding the search toward relevant data points.  Although we cannot reduce the intrinsic computational complexity of retraining itself, which is embedded in the current problem formulation, our method minimizes the number of retraining operations required, advancing the state of the art from "finding needles in a haystack" toward a more targeted and efficient approach.

---

### Official Review · Reviewer_cfFj · 2025-10-13
**Analyzing Fairness of Neural Network Prediction via Counterfactual Dataset Generation Review**

**Rating:** 4
**Confidence:** 3

**Summary:**

The paper focuses on looking at the problem of label bias in a dataset through the framework of counterfactual explanations. The method does this by involves generating a counterfactual dataset when given a particular test instance. The idea is that the counterfactual dataset will have training examples which have labels which have changed and this is done to see the model trained on the counterfactual dataset (which has different labels) will affect the prediction on the test instance.

In the counterfactual dataset generation, only a small number of samples will be chosen to change due to computational costs, and the examples changed will be dependent on specific criteria. Particularly, it would be based similarity between the neuronal activation of the test data point, and the training data points as well as the linear regression surrogate. Experiments seem to be done to seem to be done to show how many counterfactual datasets can be generated by the approach as well how efficient and meaningful the counterfactual datasets are. The related work seems to be clear and goes through the areas of literature relatively well given the space limitation. Areas of methodology and the experimental evaluation was difficult to read and can be made clearer, but generally good clarity and presentation of the work.

The paper seems novel as it seems to be to the first one to introduce the concept of counterfactual dataset as well as novel way to generate counterfactual datasets. Additionally, the work focuses on the problem of label bias in neural network mdoels which is a important issue to tackle.

**Strengths:**

- The idea of counterfactual dataset generation is novel and uses counterfactuals in a different way as what has been done before.
- Tackling the issue of label bias which is an important issue for the deployment of neural network models.
- Introduction makes it clear how we are using counterfactuals in a different way than we usually use counterfactual.
- Generally good notation, through the work,
- Clear related work section

**Weaknesses:**

- Had difficult reading secition 4.1, is there any significance with y' being in bold compared to the rest of the notation.
- Could explain the baselines a bit more, i assume you are choosing data points at random and then flapping the labels, and choosing the data points which have the lowest l2 distance to the test data point and flipping that?
- Results for RQ4 seems quite brief, potentially elaborate more.
Why does fig 3(a) and (b) not have any of the baseline’s present?
- Is there any other ways of quantiying bias which you could compare with?

**Justification:**

The paper has good clarity and presentation and is tackling an important issue in the field. There seems to have good experimental evaluation though the approach seems to be the first of its type and I am unsure iif there are other ways to qunatify bias, which could be useful to compare with.

---

> ### Author Rebuttal · Authors · 2025-10-22
>
> We thank Reviewer cfFj for their thoughtful response.  Here are our answers to the questions.
>
> * In section 4.1, the reason y \in D and y' \in D' are bolded is to distinguish these sets of labels from y = f(x) and y' = f(x'), which are single labels predicted by the function.
> * Yes, your understanding is correct.  The "Random Sampling" baseline chooses data points at random and flips their labels, and the "L2 Distance" baseline chooses data points with the lowest L2 Distance to the test data point and flips their labels.  Do note that we ensure that only the training data points that pass through the test filter \psi are considered, even for these baselines.
> * Fig. 3(a) and 3(b) actually do have the baselines present.  For 3(a), we present the ground truth data point instead, because the other baselines fail to discover any CFD (hence no data point to report).  For 3(b), we do report the Random and L2 data points in orange and pink respectively.
> * With regards to RQ4, we highlight how the test data points for which we discover CFDs are not simple "easy" boundary cases that can be flipped trivially.   We demonstrate this via Figure 4, where most test cases that are successfully found with CFDs by baselines linger around 0.0 (decision boundary for binary classification), whereas our method actually contains a wide range denoted by the whiskers.  We can make this clearer in the main text through a revision upon acceptance.
> * To the best of our knowledge, there are no existing quantitative metrics or methods we can compare with.  As mentioned in the main text, fairness research mostly focuses on inference-only step, and even the few recent works that look at alternate training datasets are on simpler ML models and thus cannot be directly used for the context of deep learning.

---

### Meta-Review · Area_Chair_aXwc · 2025-10-31

**Recommendation:** Accept (Poster)
**Confidence:** 4

**Metareview:**

There is general support for the novel idea behind the paper, though several important weaknesses were identified in terms of in evaluation, framing of the experiments with regards to fairness, scalability and connection to prior literature.

While the rebuttals successfully clarified some of the concerns (and the authors promised to incoporate changes to address the related work),  the method’s practical scalability and insufficient empirical validation of fairness claims remained only partially addressed.

However, overall, we recognize the originality and potential impact of this work (despite reservations about completeness).

---

### Decision · Program_Chairs · 2025-11-05

**Decision:**

Accept (Spotlight)

**Comment:**

We recommend an oral and a poster presentation given the AC and reviewers recommendations.

A spotlight presentation refers to a poster selected for an oral highlight but not designated as a full oral presentation per the AC’s recommendation.